# Imitating Emergencies: Generating Thermal Surveillance Fall Data Using Low-Cost Human-like Dolls

**DOI:** 10.3390/s22030825

**Published:** 2022-01-22

**Authors:** Ivan Nikolov, Jinsong Liu, Thomas Moeslund

**Affiliations:** Visual Analysis and Perception Laboratory, Aalborg University, Rendsburggade 14, 9000 Aalborg, Denmark; iani@create.aau.dk (I.N.); tbm@create.aau.dk (T.M.)

**Keywords:** thermal cameras, fall detection, thermal mannequin, anomaly detection, machine learning

## Abstract

Outdoor fall detection, in the context of accidents, such as falling from heights or in water, is a research area that has not received as much attention as other automated surveillance areas. Gathering sufficient data for developing deep-learning models for such applications has also proven to be not a straight-forward task. Normally, footage of volunteer people falling is used for providing data, but that can be a complicated and dangerous process. In this paper, we propose an application for thermal images of a low-cost rubber doll falling in a harbor, for simulating real emergencies. We achieve thermal signatures similar to a human on different parts of the doll’s body. The change of these thermal signatures over time is measured, and its stability is verified. We demonstrate that, even with the size and weight differences of the doll, the produced videos of falls have a similar motion and appearance to what is expected from real people. We show that the captured thermal doll data can be used for the real-world application of pedestrian detection by running the captured data through a state-of-the-art object detector trained on real people. An average confidence score of 0.730 is achieved, compared to a confidence score of 0.761 when using footage of real people falling. The captured fall sequences using the doll can be used as a substitute to sequences of people.

## 1. Introduction

Automated security systems are becoming increasingly ubiquitous together with the growing requirements for public safety. The possibility to offload parts of the manual surveillance from people to an automated system has driven a great deal of research in better algorithms and extending them for different use cases. These use cases can be roughly separated into outdoor and indoor use. Indoor surveillance mostly focuses on the care for children, patients and elderly [1,2].

Outdoor surveillance is directed towards detecting suspicious and anomalous pedestrian behavior on streets, in airports, in libraries [3,4] and in traffic surveillance and for the early prevention of accidents [5,6]. Thermal images are also becoming more prevalent for surveillance use cases [7], and this trend is likely to continue to rise in future years due to privacy concerns [8].

Fall detection, as apart of surveillance, has become an increasingly researched field, focusing on methods for the detection and prevention of fall accidents [9]. Fall events can lead to serious injuries and negative consequences to vulnerable groups, such as small children, patience in hospital care and the elderly. It has been shown that a quick reaction is required when such events occur and, especially for the elderly, this can prevent more severe outcomes [10]. Modern deep-learning methods have been employed steadily in tackling the problem of fall detection [11,12,13].

Such methods have proven useful in environments with single people as well as crowded areas with a large amount of foot traffic and groups of people [14,15]. Such deep-learning solutions require enough training and testing data to correctly detect falls and minimize the possibility of either missing or misclassifying important events. Most of the time, the required fall events can occur sporadically or would require certain conditions. This requires data gathering for these events that either depends on acting these scenes out in real life with volunteers [16,17] or on creating simulated videos and images [18,19].

Both of these have their pros and cons; however, generally computer simulations cost less time and resources and can produce higher amount of variations but are limited to scenarios that can be convincingly simulated. For scenarios that cannot be easily simulated, acting them out in real life is left as the only possibility. A problem that can arise in these cases is if the scenario is too dangerous or too complicated for repeated testing. In these cases, mannequins or crash test dummies can be used instead [20,21]. Ensuring that the used mannequin properly represents a human is then required. This means that the shape, size, appearance, weight, movement and interactions with the environment of the dummy need to be as close to that of humans as possible.

This paper focuses on falls happening in outdoor environments. Detecting falls into water is required as part of drowning prevention surveillance applications. Every year an estimated of 236,000 people drown around the world [22]. Compared to other types of fall events, falls in water are not that widely studied, and datasets and methods for capturing data are not thoroughly documented. Most of the research is also directed towards specific more-isolated use cases, such as drowning prevention in swimming pools [23,24] or falls from boats [11], and only some look into warning systems for every day surveillance [25,26].

As drowning events can happen around the clock and systems need to be able to register them independent of weather or lighting conditions, we choose to focus on thermal data. Thermal images are more robust to environmental changes compared to RGB and can also provide better anonymity preservation, as facial features and clothing cannot be easily recognized. We investigate the use of low-cost dolls for gathering training and testing data for fall detection using thermal images. We look into the requirements for designing a simple doll and separate these requirements into two main categories—appearance and motion.

As we focus on generating thermal data, the appearance requirements include not only the shape of the doll but also the need for it to provide a temperature distribution similar to the one from real people. The motion requirements revolve around ensuring that the construction falls in a believable way, similar to a human. An overview of the investigation presented in the paper is given in Figure 1.

To ensure that both categories of requirements are satisfied, a number of tests were conducted both in indoor laboratory settings as well as outdoor. We show that a simple inflated clothed doll can exhibit human-like temperature distribution in different body parts, such as the head, torso, hands and legs. Furthermore, this temperature distribution can be kept for a sufficiently long time in both warm and cold weather.

Fall data were also produced with the use of the doll in an outdoor environment and compared to data produced by human volunteers. The motion of both the doll and the volunteer were estimated using optical flow, which showed minimum deviations between the two. Finally, the captured doll footage was ran through a pedestrian detection algorithm and the results compared to the ones from real human footage. The detection confidence between the two was very similar with only a 4% difference.

The main contributions of the paper revolve around proving the usability of an off-the-shelf doll with for outdoor fall thermal data gathering. They can be summarized as:Experimentally verifying the visual representation of a doll construct for thermal imaging.Analyzing the motion representation of the doll for falling visualization.Presenting a real world use case for the captured thermal footage for pedestrian detection.

This work was created as a step-by-step guide to how a data-capturing system for emergency situations can be developed and verified. Its main use is for speeding up the initial process of capturing training and testing data for falling and drowning accidents, which normally require a great deal of overhead in both equipment and human-power. We show that the proposed doll system can be a viable alternative for data capture to other more complex and expensive systems [27,28,29] for the initial stages of building deep-learning-based fall detection systems.

The captured fall video dataset is provided as part of the publication—https://www.kaggle.com/ivannikolov/thermal-mannequin-fall-image-dataset (accessed on 25 October 2021).

## 2. Related Work

In this section, we discuss the related work in three main directions. Initially automated fall detection is discussed as a research area with different sensors and deep-learning techniques employed for different use cases, such as automated indoor and outdoor CCTV surveillance for anomaly detection, retirement homes and kindergarten surveillance as well as medical facility patient fall detection. All of these algorithms are shown to have the same requirements for a large amount of training and testing data.

This is why the next part of the related work discusses the ways mannequins are used for data capturing when it is hard to reproduce or dangerous scenarios are required. Finally, as we want to use thermal data for an uninterrupted 24-h fall detection, we focus on specifically thermal mannequins. This is required, as ensuring a mannequin gives off the same thermal signature as a real human is a non-trivial task that can significantly complicate the capturing setup.

### 2.1. Fall Detection

Fall detection systems can be roughly separated into indoor and outdoor use. The indoor use cases mostly focus on healthcare systems for children, the elderly or medical patients, while outdoor cases are connected to surveillance and accident prevention. Indoor fall detection systems can rely on using additional sensors and hardware for monitoring. These can be sensors mounted somewhere in the surroundings and used for surveying, such as radar [30] or Dopplers [31], or sensors directly worn by people, such as smart watches [32] or accelerometers [33].

These sensors can produce very precise time series data of a person’s position, motion and state but cannot be easily used in outdoor environments. Other possibilities are to use cameras for detection of falls. Algorithms rely on classical approaches, such as Gaussian Mixture Models and contour-based template matching [34], pose estimation and SVMs [35] and optical flow oriented histogram analysis [36]. Deep-learning techniques, such as CNNs for activity prediction and skeletal pose extraction [16,37], LSTM [1,2] and autoencoders together with transfer learning and feature residuals [17,38,39] are used on RGB, depth and thermal data.

Outdoor fall detection needs to be feasible in non-static environments with changing backgrounds, light conditions, density and clutter. This necessitates more robust approaches combining video motion detection and temporal features using three dimensional LSTM and ConvLSTM [12,40], extracting depth from single images using Markov Random Fields and human detection using particle swarm optimization [41] or body posture analysis using two-branch multi-stage CNNs [42].

Outdoor fall detection can also be directed towards detecting falls from heights, such as ships using image patch clustering and HOG features [11] or falls in harbor fronts using optical flow [26] or convolutional autoencoders and YOLOv5 object detection [43]. All these algorithms require many examples of falls, which cannot be always captured easily and with a high level of reproducibility. Mannequins can be used in many of the cases to gather enough synthetic data of falls.

### 2.2. Mannequins in Data Capture

Using mannequins and dummies for gathering testing data has been an important part of the research and development in many fields. In the automotive industry, test dummies are used for gathering data from crashes to make vehicles safer [44,45]. In medicine and the healthcare industry, mannequins are used for simulating emergency situations, such as falls of senior citizens [21] and medical patients [20,46], through the use of both camera systems and wearable sensors.

Training doctors in performing diagnosis [47] and first responders on proper CPR techniques [48] are also areas where mannequin testing data are widely used. Mannequin torsos and heads are also regularly used in testing audio wave propagation and tuning devices [49,50]. The flexibility of using mannequins provides the possibility to capture widely varying data for people’s movements, poses and interactions.

Such data can either be immoral to capture with people [51] or require a considerable time investment [52,53]. Mannequins are also used in robot vision, where obstacle avoidance and interaction with humans are required, both for generating RGB, depth or thermal images as well as point clouds [54,55].

### 2.3. Thermal Mannequins

The use of mannequins in scenarios requiring the capture and evaluation of thermal data necessitates the implementing specialized parts to simulate a thermal signature. Depending on the use case, either separate body parts, such as heads, torsos and feet, are produced or full body male, female or child representations [56]. All thermal mannequins used in research contain complicated systems required to produce visuals and readings close to those of humans—multiple heat production zones [57], sweating [58,59], soft tissue skin reactions [60], movement mechanisms [61] etc.

This becomes even more evident in for example work focusing on analyzing human thermoregulation using mannequins [61,62,63]. These mannequins are mostly suited for laboratory testing because of their size and required sensors and actuators connected to them. For outdoor environment testing, rescue mannequins and dummies can be used [27,28,29]; however, they still have the problems of high costs and being built with specific use cases. For generating thermal images of falling in water in outdoor environments, a simpler, lower-cost and easier to transport solution is required.

## 3. Capturing Cameras

Two cameras are used for the experiments in this paper. The experiments done on the harbor front use a Hikvision DS-2TD2235D-25 thermal camera [64], while all other experiments use an AXIS Q1921 thermal camera [65]. As the Hikvision is pre-installed to monitor the harbor for safety by the city municipality, the AXIS is used as a more mobile alternative as it provides internal parameters close to the Hikvision. Both are long-wavelength infrared (LWIR) cameras, which produce 8-bit grayscale images of relative temperature. The specifications of the two cameras are given in Table 1.

Both cameras capture video with a resolution of 288×384 and 24 frames per second and have comparable lens optics at 25${}^{\circ }$ mm for the Hikvision camera, versus 19${}^{\circ }$ mm for the AXIS camera. The Hikvision contains an additional RGB sensor, which is not used for the purposes of this paper. The Noise Equivalent Temperature Difference (NETD) specification of the two cameras differs with the Hikvision camera having a NETD < 50 milli-Kelvin, while the AXIS camera is rated at NETD < 100 milli-Kelvin.

The NETD is a measure of the size of difference between thermal points the camera can distinguish with a smaller NETD specifying better contrast differentiation. As we use the AXIS camera for testing the thermal visualization, we speculate that, if the doll can be detected with the camera with the worse NETD rating, then it should be detected on the one with the better. The Hikvision camera uses a vanadium oxide uncooled image sensor and has working wavelengths between 8 and 14 μm, while the AXIS camera gives no specification for the image sensor, except that it is also uncooled and the working wavelengths were shown to be between 8 and 13 μm in [66].

## 4. Thermal Doll Design

As seen in Section 2.3, current state-of-the-art mannequins used for thermal data collection are expansive, hard to transport and heavy. This makes them not suitable for use in outdoor tests, especially when the falls are into water. To address this, we selected a simple air filled rubber doll as a basis of the design. This provides a human-like shape, the rubber exterior makes the process of drying off easier, and it can be easily inflated with an air compressor or pump. The height of the doll is 1.6 m, and its weight after being fully inflated is 1.5 kg.

As the doll will be thrown in a harbor, conventional heating solutions, such as electrical pads, vests, gel thermal pads etc. would be unusable, as the combination of sea water, dirt, seaweed and low temperature would easily degrade and destroy them. A simpler solution was thus selected where water is boiled and put in sealed thermoses. Before each experiment, the water is poured onto the doll.

As there will not be a solution to continuously provide heat to the doll’s exterior, and it is made out of rubber, which has bad thermal conductivity and retention, a layer of clothes is required. Each part of the doll’s body that should be detected by the thermal cameras is clothed, using polyester clothing consisting of a tracksuit, sweatshirt with a hood, gloves, socks and winter hat. The clothes are chosen as the material would keep the heat from the applied water, without losing their shape or shrinking.

To make the doll behave closer to a human when thrown, four training ankle weights are strapped to it—one on each hand and leg. Each weight is 2 kg, boosting the overall weight of the mannequin to 9.5 kg. Finally, because of the additional weight strapped to it, together with the weight from the wet clothes, the mannequin requires additional structural support. Aluminum tube supports are made for each leg and connected to a main structure at the lower back of the doll. A heavy base stand is also made, with slots for the leg supports, so the doll can be set up standing for the easier pouring of hot water onto it. The final construction can be seen in Figure 2.

## 5. Doll Thermal Appearance

To test the thermal visuals of the doll after hot water is poured over it, two experiments were conducted. The first one aimed to compare the temperature of different body parts of the doll to the temperature of humans in the same environment. This would show that the visual representation of the clothed doll would be close enough to a human, when thrown in the water. The second experiment would test the temperature change of the clothed doll over time. This is necessary as no persistent source of heat is applied, and it is expected that the initial heat from the hot water would dissipate over time, especially in colder weather. For both experiments, we used the AXIS Q1921 thermal camera.

### 5.1. Comparing Temperature between the Doll and Real People

The first test was designed to determine if the clothed doll would exhibit temperature readings similar to those seen in real humans after the clothes were positioned on it and the hot water has been poured on it. A static laboratory environment was chosen for this test so that any possible effects of environmental variables can be minimized. For this test, we took inspiration from the comparison between human and mannequin thermal visuals presented in the research by [63,67], together with the separation of the body in heat zones for the detection of visuals of different facial expressions presented in the work by [68].

We captured the thermal visuals of 11 volunteers, as well as the doll from four distances—1, 2, 3 and 4 m. We choose the nearer distance as only a very small part of the participants could be seen at the farther distance due to the constraints of the laboratory. We separated six thermal zones—head (H), body (B), left (LA) and right (RA) arm and left (LL) and right (RL) leg. Example of these zones on a participants and the doll can be seen in Figure 3.

The AXIS camera captures uncalibrated thermal images, which change their intensity depending on the real maximum temperature of the captured object and surrounding environment. This can cause variations in the thermal images. To minimize these variations, a simple scaling step was included so that all captured intensities were transformed to real world temperatures.

To do this, we used an infrared thermometer to measure the temperature in Celsius in the body center of each participant and the doll. We then calculated the average intensity of a square of pixels from the captured thermal images in the center of the body of each participant, and the doll at each captured distance (seen as blue squares in Figure 3a,b).

The ratio between the thermometer measurements and the average intensities can then be used to scale the intensities of all other pixels of each image to real world temperature. Figure 4 show the average temperature value for each body part of the participants compared to the doll’s body part temperature. The doll had an overall higher temperature in the leg zones and lower temperature in the head zone. The readings were relatively stable between the three distance measurements. The results show that the clothed doll exhibited overall temperature readings close to the ones from humans and that it can be used as a visual replacement of a human.

### 5.2. Doll Temperature Change over Time

It was shown that the doll’s temperature representation after hot water has been poured over it closely resembles the ones from real humans. The change over time of the doll’s temperature needs to be studied, as it is not dynamically maintained over time. It is expected that, without a constant source of heat, the initial measured temperature would decline steadily and that the speed of the decline would depend on the environment in which it is. Knowing how the temperature would change over time is necessary so that enough time is given for performing the fall experiments and generating data while the temperature remains optimal.

We measured the change of the doll’s temperature from the time the hot water was poured until the difference between it and the background becomes small enough that it would be hard to distinguish it. In our case, this difference should be at least 5 ${}^{\circ }$C. We measured this change in three different scenarios—outdoor in cold weather, outdoor in mild weather and indoor. For the temperature readings, we calculated the average temperature in Celsius using the same scaling presented in the previous section. The results from these measurements are given in Figure 5.

The cold weather scenario was captured at an environmental temperature of 0 ${}^{\circ }$C, the warm weather scenario was captured at environmental temperature of 17 ${}^{\circ }$C and the indoor scenario at a temperature of 24 ${}^{\circ }$C. In both outdoor scenarios, there was wind present. In cold weather, the temperature of the doll changed at an average rate of 0.12 ${}^{\circ }$C/s at 0.06
${}^{\circ }$C/s for mild weather and 0.03
${}^{\circ }$C/s indoors. In the outdoor scenarios, the wind lowered the temperature faster, with the cold weather contributing to an even faster decline. On the other hand, the difference between the environment and the doll was much larger in cold weather, compared with in the warmer weather and indoors. In all cases, there should be at least 3 to 4 min for performing a fall. This would require a re-application of hot water after each fall.

## 6. Fall Motion Comparison

To obtain better insights into the fall behavior of the created doll construction, it was compared to real people falling in the harbor front. For this test, the Hikvision DS-2TD2235D camera was used, mounted in the outdoor environment where it will be used—overlooking a harbor front in the city of Aalborg, Denmark, as seen in Figure 6.

We used five videos captured as part of the publication by [26] of five real people falling in different ways—from stationary position, while walking and while running. We then captured videos of the doll in the same place. All in all, 22 videos of falls were captured in different scenarios—being moved along the edge and pushed in, falling in without exterior help, being thrown in, being kicked in, etc. Example frames from the people and doll tests can be seen in Figure 7a,b, with the volunteer and the doll shown with a red arrow.

First, the number of frames from the person starting falling to hitting the water was measured. The average number of frames between the recorded participants was 23. This was compared with the average number of frames the doll falls, which, in our case, was 29 frames. The longer fall times can be explained as the doll was pushed close to the harbor wall, while the volunteers were required to jump farther away from the wall for safety reasons. The doll was also affected more by the strong wind than the volunteers, because of its weight. To compare the motion of falling for the doll and the volunteers, optical flow was used as is widely seen in the literature.

We first calculated the optical flow from the videos and calculated the maximum vector magnitude in a square area on the edge of the harbor, where volunteers and the doll were falling for each frame. This area was manually selected and annotated to minimize the change of errors. This gave us a time-wise signal of the maximum detected position change and simplified the comparison, by eliminating variations in their orientation. Examples of the changes of the optical flow magnitude for the five volunteers can be seen in Figure 7c and for the first five doll experiments at Figure 7d. Clear peaks can be seen with a cutoff when hitting the water, showing that a fall can be detected in both cases.

For easier comparison, the peaks were detected, and the signals were registered so that all the peaks overlap. To compare the volunteer and doll results, we padded them to have the same length and calculated their average. The mean signal for each can be visually compared in Figure 8. The maximum peaks of each, which were 12.54 pixels/frame for the people and 10.48 pixels/frame for the doll show that the captured movement was comparable between the two.

To further compare the captured optical flow magnitude signals, we used dynamic time warping (DTW) [69]. The technique is useful for comparing time series signals, which are not perfectly aligned and with different lengths, when simply calculating an Euclidean distance between the them would not work. DTW can be used in this case as both magnitude signals have the same indexing and the same sampling. Dynamic time warping calculates the difference between the current, previous and next points in both signals and uses these as costs, selecting the minimum ones. This way, even if the two signals change with different speeds, the correct indices can be used for comparing them.

As we do not have a ground truth, we use the average volunteer optical flow time signal as one. We calculate the DTW distance between the average volunteer ground truth and each of the 22 doll optical flow signals. For comparison, we also calculate the DTW distance from the average ground truth to each of the volunteer signals. This will give an idea how the doll fall behavior compares to variations of the captured volunteer fall behavior.

The average DTW distance between the ground truth and the doll tests was 60.16, while the average DTW between the ground truth and the volunteer tests was 33.84. This shows that, even though the peak values of the volunteer and doll falls were quite similar, the overall trajectories differed. This can also be explained with the difference in weight and the safety requirement that the participants jump farther away from the harbor wall, while the doll is both thrown farther and pushed close to the wall.

## 7. Doll Detection Comparison

To show how the doll fall videos compare to real human ones from the point of view of a real computer vision application, we used an object detector trained to detect pedestrians. For this, we chose the YOLOv5 model [70] as it provides state-of-the-art performance and has been proven robust on thermal data [71,72]. The model was trained on data from the multi-seasonal LTD dataset [73] so that it had the best possible chance of detection. The model was trained for 200 epochs with a learning rate of 0.00075 on a NVIDIA RTX2080Ti graphics card.

Both the volunteer and the doll videos were separated into frames, and only the ones between jumping off the harbor edge and hitting the water were selected. This was done to limit the test to only specific instances connected to falling in the water and the behavior leading to the fall. The frames after hitting the water were also skipped. In the case of the volunteer videos, they stay mostly submerged and swim out of the field of view of the camera. The doll looses heat very fast after hitting the cold water, making it hard to distinguish it from the background.

The confidence score for the doll and the volunteers for each frame was saved, and an average confidence score was calculated for each of the videos. From these scores, an overall average score was calculated for all doll and the volunteer videos. In addition, the percentage of frames in which the volunteers and the doll were detected in each video was also calculated. The third calculated value shows how many frames before the person or doll hits the water and the object detector loses track. This is important as both the volunteers and the doll change their orientation and shape in the air by bending their limbs, thus, making it harder for YOLOv5 to detect them.

All three are given in Table 2. The average detection confidence scores of the volunteers and doll videos are very similar as well as the number of detected frames. On the other hand, in 3 of the 22 doll videos, the YOLOv5 model could not detect the doll. Upon further inspection two of these videos show the doll being thrown at a angle that is closer to horizontal to the camera view, while, in the third video, the doll has not been sufficiently warmed up, making it difficult to distinguish. Problem frames with a horizontal view and the doll not warm enough are shown in Figure 9a,b.

Some interesting observations can be made from the last row of Table 2. For the doll videos, the model loses track of an average of two frames before the it hits the water and a maximum of four frames. For the volunteer videos, the model loses track an average of one frame before the volunteers hits the water and a maximum of two. This can be attributed to the fact that once the doll is thrown or pushed, its hands and legs move very little, even with the added weights, making the overall shape less human-like (Figure 9c).

Finally, the lower percentage of frames that the doll was detected, compared to the volunteers can be explained with the fact that doll is being carried or propped up on objects before falling in the water, which can result in occlusions and losing detection (Figure 9d). Examples of successfully detected doll frames from different clips, together with confidence scores can be seen in Figure 10. It can be seen that the doll was detected with similar confidence as the other people present in the image.

The major strengths of the proposed work are the straightforward and easily replicable pipeline for developing the doll and verification. We show that, even though the physical measurements of the doll are not directly comparable to a human, the captured thermal data and movement trajectories are sufficiently convincing that they can be used as data for a deep-learning system. The proposed doll can be also easily deployed in “in-the-wild” scenarios, with minimal logistical overhead. Having demonstrated that thermal data can be captured from the doll extends its usability beyond only RGB data capture that most of the reviewed related work was using mannequins for.

## 8. Conclusions and Future Work

Detecting falls into water is an important step for preventing drowning accidents. Drowning is a major public health problem that can occur at any point of time. This makes it necessary for automated surveillance to be able to detect and signal accidents as soon as they happen. Footage of fall accidents like these is captured rarely, making it necessary to synthetically generate enough diverse data for the successful training and testing of automatic systems. Using real people can pose a health and safety risk, and this brings ethical concerns about preserving anonymity. On the other hand, generating data using simulation and deep-learning methods can provide imperfect results with the necessity of post-processing steps.

This is why, in this paper, we demonstrated that a rubber air-filled doll can be used for generating thermal image fall data in outdoor scenarios. By using off-the-shelf clothes and hot water, the doll can achieve human-like thermal properties and maintain them for an extended period of time in different weather conditions. Throwing and pushing the doll off a ledge also achieved similar movement vectors to real people jumping.

Finally, we showed that a YOLOv5 object detection algorithm trained on people can detect the thermal signature of the doll with confidence close to the one for detecting real people. As this is the first dataset focused on generated thermal fall data and the detection of such accidents is crucial for automatic surveillance systems, we made the dataset available online so that others can benefit from our data.

The proposed solution can be also applicable to other domains, such as generating traffic accident data, as well as indoor anomaly detection use cases, such as generating fall data for the elderly and children.

We encountered certain limitations in the proposed work. The rubber doll, even with the added supports and weights, had an overall weight of 9.5 kg, which is far from a regular human weight. This helped with making the system more easily transportable but lowered the precision of the captured fall movement data. The simple method of raising the temperature of the doll can also be viewed as a positive feature and a limitation. The inability to maintain a stable temperature for long periods of time would require additional equipment for performing repeated precise temperature experiments.

Potential improvements for the proposed solution can be done by addressing the difference between the weight and articulation of the doll and a real human.To address this, adding a weighted vest is proposed, together with 3D printed joints for the arms and legs of the doll. This would give the possibility to add more weight—up to 15 more kilograms, making the full weight of the doll up to 25 kg and matching other off-the-shelf doll solutions with the added benefit of flexibility. This would also make the additional burden on transportation and setup less problematic.

For extending the thermal signature of the doll across time, we propose adding heating thermal pads to the clothing. These pads would be connected to isolated thermal sensor pads and an Arduino or Raspberry-Pi in a water-tight case. In this way, data from the clothes temperature of the doll can be sent through a low-power Bluetooth or radio signal to a monitoring station, and, when necessary, the pads can be changed or more heating ca be applied.

## Figures and Tables

**Figure 1 sensors-22-00825-f001:**
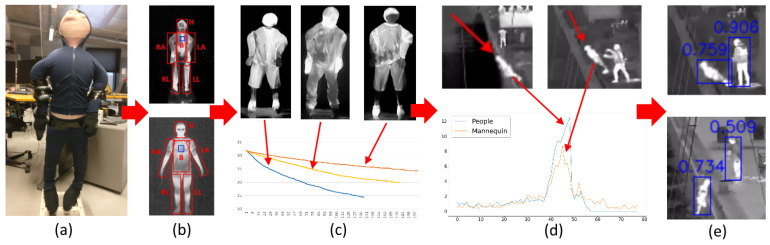
Overview of the proposed system and the investigation done in the paper. First, a human-like figure is designed with appropriate weights and support (**a**), followed by comparisons between the thermal signature of humans and the doll (**b**) and the stability of the signature through time (**c**). The fall motion of the figure is then compared to those of humans (**d**). Finally, the usability of the generated data is verified using a pedestrian detection algorithm (**e**).

**Figure 2 sensors-22-00825-f002:**
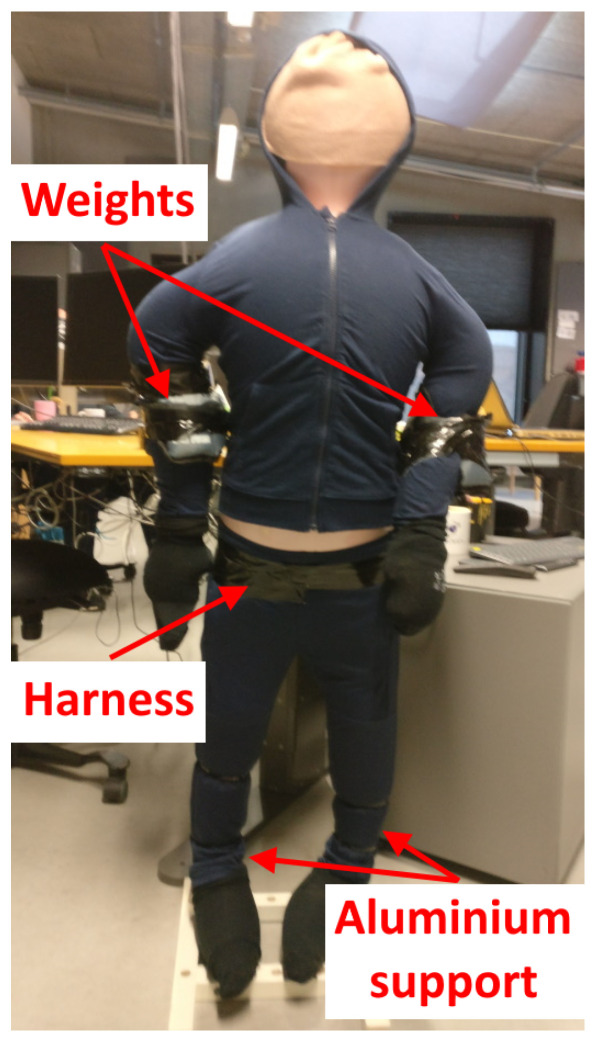
View of the created low-cost doll, together with clothes and ankle weights.

**Figure 3 sensors-22-00825-f003:**
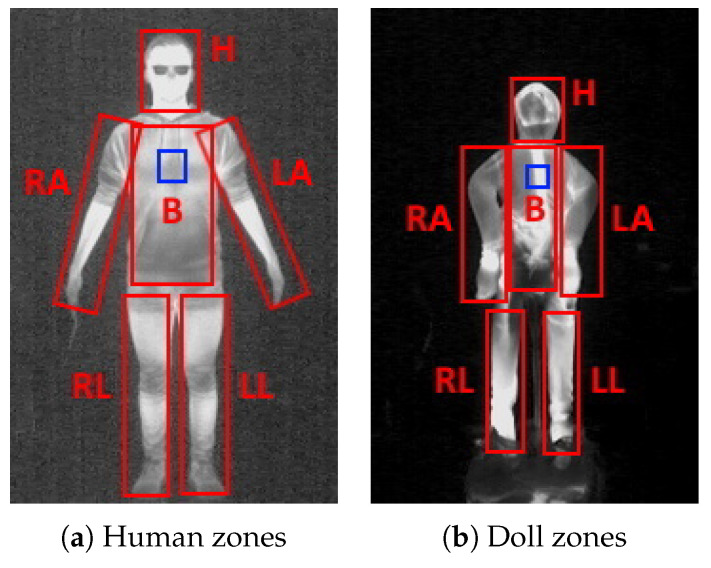
Visualization of the six thermal zones—head (H), body (B), left arm (LA), right arm (RA), left leg (LL) and right leg (RL) on one of the volunteers and the doll. Blue squares show the central pixels used for calculating the thermal image calibration.

**Figure 4 sensors-22-00825-f004:**
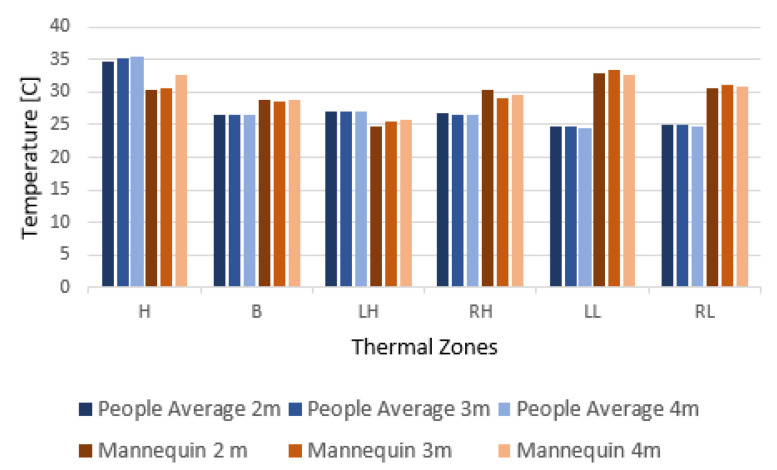
Average temperatures of the volunteers’ thermal zones, compared to the ones from the doll captured from 4, 3 and 2 m, respectively.

**Figure 5 sensors-22-00825-f005:**
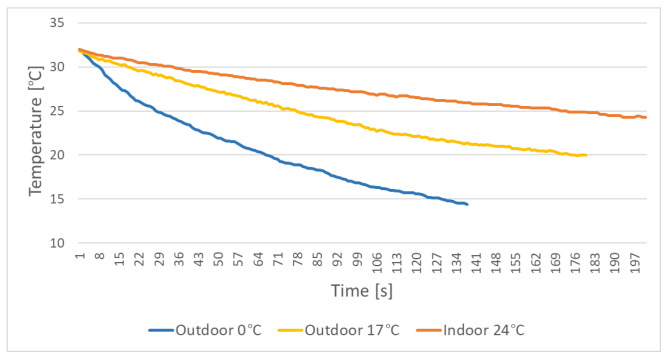
Measured doll temperature over time in three different scenarios—outdoor in cold weather at 0 ${}^{\circ }$C, outdoor in mild weather at 17 ${}^{\circ }$C and indoor at 24 ${}^{\circ }$C.

**Figure 6 sensors-22-00825-f006:**
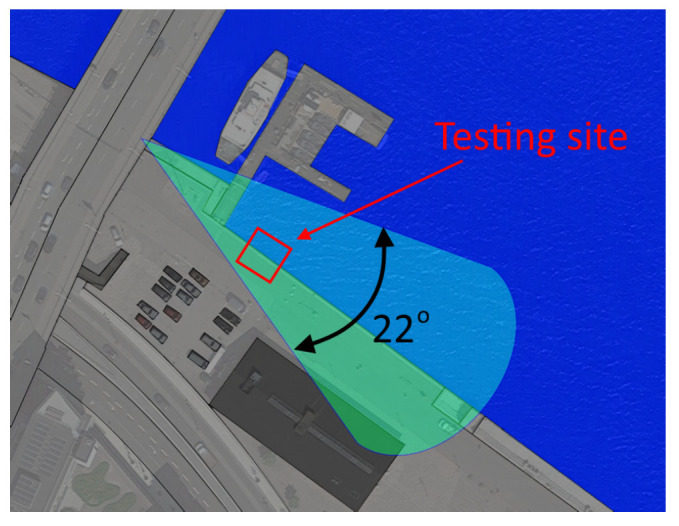
The area and field of view of the Hikvision DS-2TD2235D camera used for the outdoor experiments.

**Figure 7 sensors-22-00825-f007:**
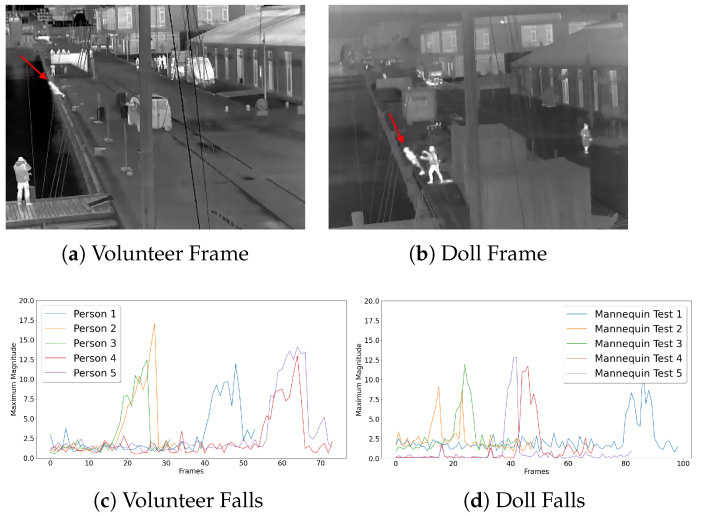
Examples of frames from a volunteer (**a**) and a doll (**b**) falling, shown with a red arrow, together with the maximum magnitudes of the optical flow vectors of five videos from the falling person and doll (**c**,**d**). The clear spike when they fall can be seen. The spike is in different positions as it took different amounts of time before the fall.

**Figure 8 sensors-22-00825-f008:**
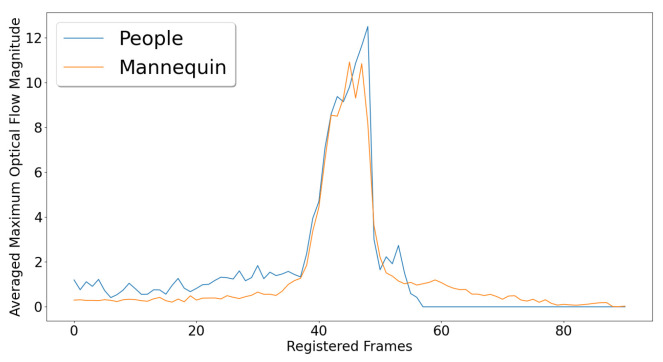
The averaged maximum optical flow magnitudes between the different people and doll falls. Before averaging the fall data, the peaks were registered.

**Figure 9 sensors-22-00825-f009:**
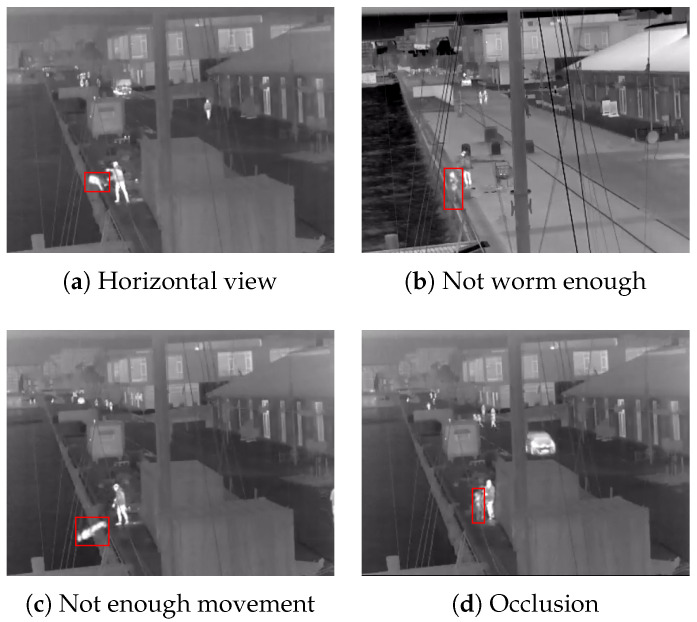
Problems seen with the doll clips. The object detection can fail before the doll is thrown if it is not warm enough (**b**) or is occluded by a person or object (**d**). After the doll is pushed or thrown, the object detection can fail if the body is horizontal and cannot be seen by the camera (**a**) or not enough movement is present in the limbs (**c**).

**Figure 10 sensors-22-00825-f010:**
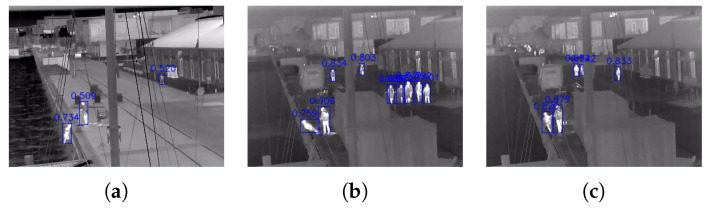
Examples of successful doll detection from the YOLOv5 model together with the confidence scores. Both in scenarios where there are no other people (**a**) and many other people (**b**,**c**), the doll is detected before it hits the water.

**Table 1 sensors-22-00825-t001:** The two thermal cameras used in the experiments. The main testing camera is the Hikvision DS-2TD2235D, while the AXIS Q1921 is used for laboratory testing. The Hikvision additionally contains an RGB sensor, which is not used in the experiments for this paper.

	AXIS Q1921	Hikvision DS-2TD2235D
Resolution	384 × 288	384 × 288
Image sensor	Uncooled	Vanadium Oxide Uncooled
Response Waveband	8–13 μm	8–14 μm
FoV	19	25
Output	8-bit grayscale	8-bit grayscale
NETD	<100 milli-Kelvin	<50 milli-Kelvin

**Table 2 sensors-22-00825-t002:** Average YOLOv5 detection results for the volunteers and doll videos. These consist of confidence scores for the detected instances in the frames, percentage of frames in which the volunteer or doll have been detected and the number of frames before hitting the water, where the volunteers or doll were not detected.

Results	Volunteers	Doll
Avg. Confidence Score	0.761	0.730
Avg. Detected Frames [%]	77.7	62.1
Avg. Lost Frames before Water Hit	1	2

## Data Availability

The dataset of doll construction falls and the code for reading and analyzing it are provided in the link—https://www.kaggle.com/ivannikolov/thermal-mannequin-fall-image-dataset (accessed on 25 October 2021).

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
