# Peer review of "Imitating Emergencies: Generating Thermal Surveillance Fall Data Using Low-Cost Human-like Dolls"

_sensors, 2022, doi:10.3390/s22030825_

Round 1
Reviewer 1 Report
I’m not sure the authors aptly named a used rubber doll filled with air because the thermal manikin is used to evaluate the thermal insulation of clothing. I find that air filled rubber dummy weighing only 1.5 kg cannot be used for generating thermal image fall data in outdoor scenarios. It is unclear how "... by using off-the-shelf clothes and hot water, the mannequin can achieve human-like thermal properties and keep them for an extended period of time in different weather conditions." The authors state "..he dummy it is dressed in normal off the shelf clothes consisting of a tracksuit, sweatshirt with a hood, gloves, socks and winter hat ...". Different textile materials have different water absorption capacity as well as different thermal properties. In general, wet clothes in the cold cause great thermal discomfort.
What is the purpose of "... Before each experiment the water is poured onto the clothed mannequin ..." Did the authors think about how to measure and maintain a uniform core body temperature for a long time during the experiment?
Author Response
The authors would like to thank the reviewer for taking the time to go through the paper and pointing out problems and commenting on shortcomings. We tried to take these into consideration and to address the concerns of the reviewer.
First, we have expanded the Conclusion and Future Work section to address the shortcomings of our current solution and to further elaborate on the next steps that will be taken to expand the solution and make it more easily usable. Currently, there is no easy way to judge the temperature of the mannequin other than measuring it with an infrared thermometer. To address this our future work would contain thermal sensors and an on-board micro-controller like an Arduino in a water-proof case, which would give us a better overview of the temperature changes over time and when the mannequin hits the water.
Currently, the mannequin together with the added supports and weights is at around 9.5kg, which even though far from the weight of a full-grown human, can still approximate a falling trajectory for creating a training or testing data when such data cannot be captured by a human. Please see some of the fall videos as examples of that https://www.kaggle.com/ivannikolov/thermal-mannequin-fall-image-dataset. In our future work, we would like to add a weighted vest and additional supports for the hands. This would add an additional weight of around 15 to 16 kg that can be added or removed depending on the use case. We have added that to the Future Work part.
We used off-the-shelf clothes in an effort to make the solution as easily replicable as possible. Each part of the system should be easy to get ahold of, as we have seen that building such a system by buying thermal manikins or crash-test dummies can become prohibitively expensive. We have further elaborated that in a new part of the introduction that we specify the main goal of the paper. We absolutely agree with the reviewer's opinion that wet clothes in cold weather and especially in cold water can cause thermal discomfort and even worse outcomes. We use the simple solution of pouring hot water on the clothes layer so we can achieve a thermal signature that is comparable to a normal thermal signature to a human as seen in Figure 3. We have expanded that part of the paper to give a better overview of why we use these clothes. We have seen of course that once the mannequin falls in the harbor water which in our cases was around 7 degrees this thermal signature falls very fast and requires the reapplication of hot water after a couple of falls. This is why we believe that thermal pads can be the next step for our future research. Electrical heat vests and pads would be too dangerous as the saltwater that the mannequin falls in can erode and damage the components.
In addition, we have expanded the Related work and Introduction with additional references to related work and to the significance of why falls should be detected. We have also added our work in the context of what other work is out there and why we believe that even though still in the initial stages the paper can be useful to other researchers.
Reviewer 2 Report
This work is well presented. Author need to address the following minor comments;
1) Add the importance of this work in the introduction section. Also add the latest articles in the introduction section and update the related work.
2) The application of this work should be added with proper reference article. Also add the summary of related work.
3) What is the major strength of this work? discuss in the results section.
4) In the conclusion of this work, add the limitations of this work.
Author Response
The authors would like to thank the reviewer for taking the time to read the article and the given comments and directions to improve the overall quality of the proposed paper. Below is a point-by-point response of how we address the comments.
- In the introduction, we added a paragraph explaining the importance of detecting falls in both indoor and outdoor scenarios. We have added a reference to the importance of quick response time. As well as provided additional citations of research focussing both on scenarios where there are groups of people, as well as single people. At the end of the introduction, we have added a paragraph putting the results of our work in the context of the existing solution.
- We expanded the introduction to show the significance of the current research in the context of the limited scope of other research in the area and the difficulty of capturing data. We change the name of State-of-the-art to Related Work, to better suit the wider context of the work presented there and added a summary making the reading experience hopefully easier to follow.
- We added two paragraphs to the end of the results section commenting on the strengths of our work and the possible use cases that we see it would benefit
- We have expanded the Conclusion and Future work section to contain an expanded limitation part, as well as expanded future work with the next step we will be undertaking.
Round 2
Reviewer 1 Report
The authors did not fully answer my questions from Round 1.
They gave general answers. Please answer every single question from Round 1.
What does "..use normal normal off the shelf clothes ..." mean?
The following link is not active (Message: 404 We can't find that page.)
https://www.kaggle.com/ivannikolov/thermal-mannequin-fall-image-dataset
Author Response
First off the authors would like to apologize for the inactive link. The dataset was set to private, we have changed that, and now it should be accessible - https://www.kaggle.com/ivannikolov/thermal-mannequin-fall-image-dataset
We would also like to apologize that the provided answers were not satisfactory and would like to elaborate on them below.
- "I’m not sure the authors aptly named a used rubber doll filled with air because the thermal manikin is used to evaluate the thermal insulation of clothing." - After looking through the related work again and taking into consideration the comment of the reviewer, we saw that the use of the term "mannequin" can be confusing to readers, so we have taken the suggestion of the reviewer and renamed it to a "doll" in both the headline and the body of the paper.
- " I find that air filled rubber dummy weighing only 1.5 kg cannot be used for generating thermal image fall data in outdoor scenarios." - As mentioned the full weight of the rubber doll construction together with the weights and supports becomes 9.5 kg. We saw that even that weight can better represent a falling motion, it still causes some discrepancies compared to a human falling. We have added a discussion of this problem to the end of the Results section and Conclusion and we have given examples in the Future work part how we will try to address this by using an additional weighted vest and arm supports.
- "It is unclear how "... by using off-the-shelf clothes and hot water, the mannequin can achieve human-like thermal properties and keep them for an extended period of time in different weather conditions." and "The authors state "..he dummy it is dressed in normal off the shelf clothes consisting of a tracksuit, sweatshirt with a hood, gloves, socks and winter hat ...". Different textile materials have different water absorption capacity as well as different thermal properties. In general, wet clothes in the cold cause great thermal discomfort." - We agree that the current explanation can be confusing so we have changed it. We mention that as the doll is made out of rubber and we do not use external constant sources of heating, we only rely on an initial source by the application of hot water before each experiment. Thus we need a layer of clothing to which the hot water can adhere and that has properties to retain enough thermal heat for a period of time that each experimental throwing/fall is performed. After each throw, the hot water is reapplied to the clothes as the contact with the cold harbor water rapidly cools down the temperature. We now elaborate on the "off-the-shelf" clothes, by explaining that we have chosen polyester clothes that would retain the heat of the water and would not change their shape too much. The clothes are simple ones bought in a shop. We tested out different materials and saw that these ones provided satisfactory results. We tested the amount of time it would take the temperature of the clothes to go down after an initial application of hot water and saw that the time was enough to set up and perform a throwing/falling experiment.
- "What is the purpose of "... Before each experiment the water is poured onto the clothed mannequin ..." Did the authors think about how to measure and maintain a uniform core body temperature for a long time during the experiment?" - The purpose of the hot water poured onto the doll before each experiment is to get it to a high enough temperature that it will be registered by the thermal camera, as normally clothed or not clothed the doll had a very low temperature and could not be distinguished from the background in a thermal camera feed. As this is still the initial study into the problem we saw that as suggested we need to have a way to monitor the temperature automatically. Currently, we monitor the temperature manually using an infra-red thermometer before and after each application of hot water and after the doll has been pulled out of the harbor water. As stated in the future work we plan to create a watertight container with sensors and an Arduino to send the temperature current temperature of the doll at specific places on its body so we know when it is required to apply more hot water and to also know that we can achieve relatively consistent high temperatures for repeated experiments.